

# Construction, internal validation and implementation in a mobile application of a scoring system to predict nonadherence to proton pump inhibitors

Emma Mares-García[1], Antonio Palazón-Bru[1], David Manuel Folgado-de la Rosa[1], Avelino Pereira-Expósito[2], Álvaro Martínez-Martín[1], Ernesto Cortés-Castell[3] and Vicente Francisco Gil-Guillén[1,2]

[1] Department of Clinical Medicine, Miguel Hernández University, San Juan de Alicante, Alicante, Spain
[2] Research Unit, General University Hospital of Elda, Elda, Alicante, Spain
[3] Department of Pharmacology, Pediatrics and Organic Chemistry, Miguel Hernández University, San Juan de Alicante, Alicante, Spain

## ABSTRACT

**Background**. Other studies have assessed nonadherence to proton pump inhibitors (PPIs), but none has developed a screening test for its detection.

**Objectives**. To construct and internally validate a predictive model for nonadherence to PPIs.

**Methods**. This prospective observational study with a one-month follow-up was carried out in 2013 in Spain, and included 302 patients with a prescription for PPIs. The primary variable was nonadherence to PPIs (pill count). Secondary variables were gender, age, antidepressants, type of PPI, non-guideline-recommended prescription (NGRP) of PPIs, and total number of drugs. With the secondary variables, a binary logistic regression model to predict nonadherence was constructed and adapted to a points system. The ROC curve, with its area (AUC), was calculated and the optimal cut-off point was established. The points system was internally validated through 1,000 bootstrap samples and implemented in a mobile application (Android).

**Results**. The points system had three prognostic variables: total number of drugs, NGRP of PPIs, and antidepressants. The AUC was 0.87 (95% CI [0.83–0.91], $p < 0.001$). The test yielded a sensitivity of 0.80 (95% CI [0.70–0.87]) and a specificity of 0.82 (95% CI [0.76–0.87]). The three parameters were very similar in the bootstrap validation.

**Conclusions**. A points system to predict nonadherence to PPIs has been constructed, internally validated and implemented in a mobile application. Provided similar results are obtained in external validation studies, we will have a screening tool to detect nonadherence to PPIs.

# INTRODUCTION

Proton pump inhibitors (PPIs) are prescribed in clinical practice for the treatment of gastro-esophageal reflux disease, as well as other acid-related disorders (*Robinson & Horn,*

Corresponding author
Antonio Palazón-Bru,
antonio.pb23@gmail.com

*2003*). The indications for their use are increasing, especially in patients with digestive problems, or those who are taking a medication that may cause damage or secondary diseases such as gastritis, digestive ulcers or bleeding (*Domingues & Moraes-Filho, 2014*).

Approximately 20–42% of patients may not respond correctly to PPI therapy, which can cause gastrointestinal complications in patients using anti-inflammatory drugs (NSAIDs) (*Van Soest et al., 2007*). One of the main factors associated with the lack of effectiveness of PPIs is therapeutic nonadherence, the prevalence of which can reach up to 50% (*Domingues & Moraes-Filho, 2014*; *Henriksson, From & Stratelis, 2014*). It has also been shown that patients have lower adherence to PPI therapy when there are certain sociodemographic factors, symptoms of gastrointestinal complications, lack of understanding about taking medication or reason for prescription, adverse effects, and an inadequate doctor-patient relationship (*Sturkenboom et al., 2003*; *Fass et al., 2005*; *Hungin, Rubin & O'Flanagan, 1999*; *Dal-Paz et al., 2012*; *Lanas et al., 2012*).

To detect patient nonadherence to PPI therapy, we used the percentage of days covered by the PPI (*Domingues & Moraes-Filho, 2014*; *Henriksson, From & Stratelis, 2014*), the pill count (*Lanas et al., 2012*) or the Morisky test (*Dal-Paz et al., 2012*; *Domingues & Moraes-Filho, 2014*). The first two methods are considered objective and allow accurate determination of whether the patient is nonadherent, but are difficult to apply in clinical practice. On the other hand, the Morisky test is not as accurate as the methods mentioned above and there must be a good doctor-patient relationship (*Perseguer-Torregrosa et al., 2014*). In other words, we do not have an objective measure that is easy to apply in clinical practice and that gives us accurate results, i.e., a screening test to determine nonadherence to PPI therapy. For this reason we decided to conduct a prospective study, constructing and internally validating through bootstrapping a predictive model of nonadherence to PPI therapy using objective, easy to measure factors. To facilitate its implementation in routine clinical practice, this model was adapted to a points system and implemented in an application for the Android mobile phone operating system. Provided our points system is validated in other regions, we will have a screening tool to reduce nonadherence to PPI therapy and thus reduce possible gastrointestinal complications (*Hedberg et al., 2013*; *Jonasson et al., 2013*; *Domingues & Moraes-Filho, 2014*).

# MATERIALS & METHODS

## Study population

The study population comprised patients prescribed PPIs (omeprazole, lansoprazole, pantoprazole, rabeprazole and esomeprazole) for any cause in the towns of Elda, Santa Pola and San Vicente del Raspeig, located in the province of Alicante (Spain). This province is situated in the southeast of Spain and in 2013 had a population of 1,854,244 inhabitants. The number of inhabitants of the towns included in the study in 2013 was: (1) Elda, 54,056; (2) Santa Pola, 34,134; and (3) San Vicente del Raspeig, 55,781. The health system is free and universal. All medication prescribed by both primary and specialized care physicians is collected by the patient at the pharmacy, where all information is recorded automatically (electronic prescription).

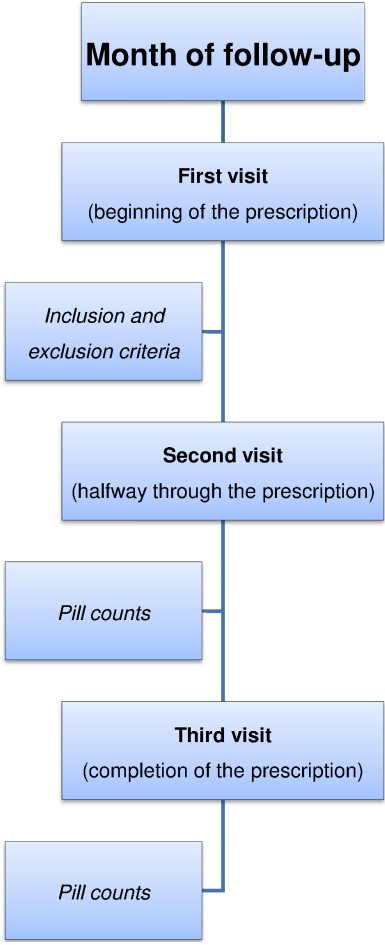

**Figure 1    Phases of our study design.**

## Study design and participants

This was a prospective observational one-month follow-up study carried out between August and October 2013, at three pharmacies in the province of Alicante (Elda, Santa Pola and San Vicente del Raspeig). All patients who visited these pharmacies during the study period to collect their prescribed PPIs were invited to participate. The PPI was prescribed by the physician for gastric protection due to use of NSAIDs, gastric problems, or possible interactions between different drugs. Since the objective of our study was to determine which patients did not adhere to PPI medication (prediction of this event), we excluded those who presented problems that prevented them from approaching the pharmacy on subsequent visits to determine whether they were taking the PPI medication correctly.

During the month of follow-up, the patients made three visits to the pharmacy (Fig. 1). To prevent the patient withdrawing tablets from the package to achieve good pill counts, as a pretext he or she was instructed to bring all packages so that quality control could be performed on the medication. This procedure was done outside of the patient's vision and only the PPI tablet count was performed. Thus information bias in the primary variable of this study, nonadherence to PPIs, was minimized.

A one-month period was chosen, even though PPIs can be prescribed for a longer period, because it was the length of time during which all tablets in the package prescribed should have been taken. Moreover, we must bear in mind that this study was conducted in pharmacies, in order to help prescribing physicians reduce patient treatment nonadherence. This is a consequence of the fact that pharmacy professionals can act on the process and outcomes of healthcare to try to improve adherence to treatment (*Foro de Atención Farmacéutica, 2008*).

## Variables and measurements

The primary study variable was therapeutic nonadherence to PPIs by the patient. To determine this variable, the ratio between the total number of tablets taken by the patient at follow-up and the total number of tablets prescribed by the doctor (obtained from the electronic prescription at the first visit) were calculated. We defined nonadherence as when the patient failed to take between 80% and 110% of the tablets prescribed by their physician (*Perseguer-Torregrosa et al., 2014*).

The secondary variables recorded at the first visit were: gender (male or female), age (years), prescription of antidepressants (yes or no), type of PPI (omeprazole or others), non-guideline-recommended prescription (NGRP) of PPIs and the total number of drugs. PPIs are indicated mainly in diseases related to gastric acid secretion. They are also used for the prevention of secondary drug gastropathies and may also be indicated in more specific pathologies that require short-term treatment (*Robinson & Horn, 2003*; *Domingues & Moraes-Filho, 2014*; *Administración de la Comunidad Autónoma del País Vasco, 2016*). All these variables are contained in the electronic prescription, which includes all information relating to drug prescriptions for each patient, along with the age and gender of the patient.

Our secondary variables were chosen with the aim of using them to explain the nonadherence. Other studies on adherence to PPIs have shown that sociodemographic factors may influence the correct taking of medication. In addition, we studied NGRP of PPIs, because the presence of gastric symptoms (main recommendation of the guidelines) (*Robinson & Horn, 2003*; *Domingues & Moraes-Filho, 2014*; *Administración de la Comunidad Autónoma del País Vasco, 2016*) is an important factor associated with adherence (*Dal-Paz et al., 2012*; *Lanas et al., 2012*). Studies on other drugs have determined that the more prescriptions a patient has, the greater likelihood of treatment nonadherence (*Perseguer-Torregrosa et al., 2014*). The variable 'prescription of antidepressants' was included in the study because there was a suspicion that a depressed patient could have greater forgetfulness when taking the medication prescribed by his or her physician. Finally, since most prescriptions contained the active ingredient omeprazole, we wanted to determine whether its use was associated with a lower nonadherence.

## Sample size calculation

As the objective of our work was to construct a predictive model (a logistic regression model), the sample size had to be based on the events-per-variable ratio. This ratio had to be greater than 10. In our study we had 99 events (patients with nonadherence), therefore with this sample size we could construct a predictive model with nine explanatory variables (*Palazón-Bru et al., 2017*).

## Statistical methods

The qualitative variables were described by calculating absolute and relative frequencies (percentages), while quantitative variables were described by calculating the median and interquartile range. Since data from three pharmacies in different locations were used, the homogeneity of the same was verified beforehand through tests based on the $\chi^2$ (Pearson or Fisher) and the median tests. After verifying the homogeneity, all data were analyzed together.

A binary logistic regression model was constructed in which the dependent variable was therapeutic nonadherence to PPIs and the independent variables were our secondary variables (gender, age, antidepressants, omeprazole, NGRP of PPIs, and the total number of drugs). Through this model, the adjusted relative risks (RR) were obtained. The goodness-of-fit of our model was performed using the likelihood ratio test, Hosmer-Lemeshow test and Nagelkerke's $R^2$. The model was adapted to a points system through the methodology of the Framingham study (*Sullivan, Massaro & D'Agostino, 2004*), which weights coefficients determining an associated score for each variable and thus can be used in routine clinical practice. This methodology has been applied to various areas of knowledge in the field of medicine (*Artigao-Ródenas et al., 2015*; *Azrak et al., 2015*; *Gutiérrez-Gómez, Cortés & Palazón-Bru, 2015*; *López-Bru et al., 2015*; *Palazón-Bru et al., 2015*; *Ramírez-Prado et al., 2015a*). After calculating the score associated with each patient, the ROC curve was calculated and the cut-off point was established as the one that minimized the square root of $(1\text{-Sensitivity})^2 + (1\text{-Specificity})^2$ (*Hanley & McNeil, 1982*). After obtaining this point, sensitivity, specificity, positive (PPV) and negative predictive values (NPV) and positive (PLR) and negative likelihood ratios (NLR) were calculated.

To perform internal validation of the points system, 1,000 bootstrap samples from the original sample were taken (random samples with replacement of the same number of elements as the original sample) (*Steyerberg et al., 2001*) and in each of them sensitivity, specificity and AUC were calculated. Thus we obtained a distribution for the three parameters, which was represented through histograms.

The type I error was set at 5% for all calculations and for each relevant parameter its associated confidence interval (CI) was calculated. All calculations were implemented with IBMS SPSS Statistics 19 R 2.13.2.

## Ethical issues

The study was approved by the Ethics Committee of the Department of Health of Elda and both data collection and analysis were conducted anonymously and encrypted. All patients gave their consent to participate in the study orally.

## Mobile application

The points system has been integrated into an application for mobile phones with the Android operating system. Download from the store (Google Play) is free. The name of this application is Nonadherence to PPIs.

**Table 1** Descriptive analysis of the patients between the three participant pharmacies (Elda, Santa Pola and San Vicente del Raspeig).

| Variable | Total<br>$n = 302$<br>$n$(%)/median(IQR) | Pharmacy 1<br>$n = 183$<br>$n$(%)/median(IQR) | Pharmacy 2<br>$n = 74$<br>$n$(%)/median(IQR) | Pharmacy 3<br>$n = 45$<br>$n$(%)/median(IQR) | $p$-value |
|---|---|---|---|---|---|
| Nonadherence to PPI | 99(32.8) | 59(32.2) | 28(37.8) | 12(26.7) | 0.439 |
| Male gender | 138(45.7) | 87(47.5) | 33(44.6) | 18(40.0) | 0.645 |
| Antidepressants | 109(36.1) | 67(36.6) | 26(35.1) | 16(35.6) | 0.972 |
| Omeprazole | 234(77.5) | 144(78.7) | 60(81.1) | 30(66.7) | 0.156 |
| NGRP of PPI | 192(63.6) | 116(63.4) | 48(64.9) | 28(62.2) | 0.955 |
| Age (years) | 70(20) | 70(20) | 67(21) | 72(16) | 0.700 |
| Total number of drugs | 5(4) | 5(4) | 6(4) | 4(3) | 0.153 |

**Notes.**

IQR, interquartile range; NGRP, non-guideline-recommended prescription; n(%), absolute frequency (relative frequency); PPI, proton pump inhibitors.
Guideline-recommended prescription of PPI: prevention of upper gastrointestinal disorders in high-risk patients (*Robinson & Horn, 2003*; *Domingues & Moraes-Filho, 2014*; *Administración de la Comunidad Autónoma del País Vasco, 2016*).

**Table 2** Multivariate analysis of nonadherence to proton pump inhibitor drugs.

| Variable | Adj. RR (95% CI) | $p$-value |
|---|---|---|
| Male gender | 1.29(0.66–2.55) | 0.456 |
| Antidepressants | 11.91(6.01–23.58) | <0.001 |
| Omeprazole | 0.87(0.38–1.96) | 0.735 |
| NGRP of PPI | 1.75(0.83–3.68) | 0.138 |
| Age (years) | 1.00(0.97–1.03) | 0.869 |
| Total number of drugs | 1.49(1.28–1.74) | <0.001 |

**Notes.**

Adj. RR, adjusted relative risk; CI, confidence interval; NGRP, non-guideline-recommended prescription; PPI, proton pump inhibitors.
Guideline-recommended prescription of PPI: prevention of upper gastrointestinal disorders in high-risk patients (*Robinson & Horn, 2003*; *Domingues & Moraes-Filho, 2014*; *Administración de la Comunidad Autónoma del País Vasco, 2016*). Goodness-of-fit of the model: $X^2 = 128.7$, $p < 0.001$, Hosmer-Lemeshow $X^2 = 7.6$, $p = 0.181$ Nagelkerke's $R^2 = 0.483$.

## RESULTS

A total of 302 patients were invited to participate during the study period. No patient refused to participate or was lost to follow-up. Our number of recruited patients was higher than the sample size calculated a priori. Of all the patients analyzed, 99 showed nonadherence to PPIs (32.8%, 95% CI [27.5–38.1%]). The descriptive characteristics of the sample (Table 1) showed a proportion of men close to half (45.7%), 36.1% had prescriptions for antidepressants, 63.6% of the prescriptions were inappropriate, the median age was 70 years and the median number of drugs prescribed was five. There were no differences between the three pharmacies analyzed in any of the variables, with $p$-values ranging between 0.153 and 0.972 (Table 1).

Table 2 shows the estimated RR through the multivariate model, which had a very satisfactory goodness-of-fit in all the tests used (likelihood ratio test, $p < 0.001$; Hosmer-Lemeshow test, $p = 0.181$; Nagelkerke's R², 0.483). After adapting to a points system (Fig. 2), three variables remained in the system (total number of drugs, NGRP of PPIs,

| Total number of drugs | Score |
|:---:|:---:|
| 1 | 0 |
| 2-3 | 1 |
| 4-5 | 2 |
| 6-7 | 3 |
| 8-9 | 4 |
| ≥10 | 5 |

| NGRP of PPI | Score |
|:---:|:---:|
| Yes | 1 |
| No | 0 |

| Antidepressants | Score |
|:---:|:---:|
| Yes | 3 |
| No | 0 |

| Total score | Outcome |
|:---:|:---:|
| ≥4 | Positive |
| <4 | Negative |

**Figure 2  Scoring system to predict nonadherence to proton pump inhibitors.** Abbreviations: PPI, proton pump inhibitors; NGRP, non-guideline-recommended prescription. Guideline-recommended-prescription of PPI: prevention of upper gastrointestinal disorders in high-risk patients (*Robinson & Horn, 2003*; *Domingues & Moraes-Filho, 2014*; *Administración de la Comunidad Autónoma del País Vasco, 2016*).

and antidepressants). Applying the scores in our patients gave an AUC near 90% (Fig. 3) and the optimal point had a value of four, that is, if the patient had a total score greater than or equal to 4 points, the test was considered to be positive. The test parameters were: sensitivity, 0.80 (95% CI [0.70–0.87]); specificity 0.82, (95% CI [0.76–0.87]); PPV, 0.69 (95% CI [0.59–0.77]); NPV, 0.89 (95% CI [0.84–0.93]); PLR, 4.50 (95% CI [3.29–6.15]); and NLR, 0.25 (95% CI [0.17–0.37]).

Bootstrap validation (Fig. 4) gave very similar parameters in the means, and the distribution of values showed very high values, as the AUC was almost always greater than 0.80, and the sensitivity and specificity above 0.75.

## DISCUSSION

### Summary
This study constructed and internally validated a points system to predict therapeutic nonadherence to PPIs. This system showed a very satisfactory goodness-of-fit both in the construction and validation. In addition, to facilitate its use by health professionals, this system has been implemented in a mobile application for the Android operating system.

### Strengths and limitations of the study
The main strength of our work is the clinical idea developed as, in an innovative way, we constructed and internally validated a prediction model to attempt to reduce nonadherence to PPIs and consequently the risk of gastrointestinal problems (*Jonasson et al., 2013*; *Domingues & Moraes-Filho, 2014*). Second, we corroborated the validation with 1,000 samples and in all of them we obtained very high sensitivity, specificity and AUC values,

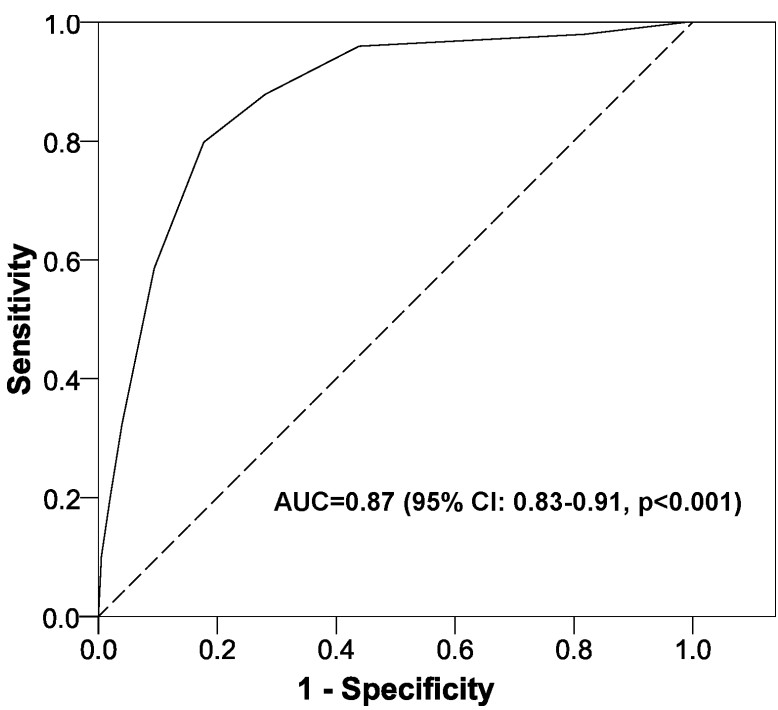

**Figure 3** **Area under the ROC curve for the scoring system to predict nonadherence to proton pump inhibitors.** Abbreviations: AUC, area under the ROC curve; CI, confidence interval.

giving greater validity to our results. Finally, using a mobile phone application in the world of technology could help implement our prediction model in clinical practice, because in just a few seconds the doctor can know if the patient is at risk of nonadherence to PPI medication.

With regard to selection bias, all patients during a specified period of time were included and there were no differences in the three pharmacies analyzed. Information bias was minimized, as our source of information (electronic prescriptions) collects all information used in our study accurately and nonadherence was assessed with an objective method (*Perseguer-Torregrosa et al., 2014*). Confounding bias was minimized through the use of a multivariate logistic regression model. Also, nonsignificant variables remained in the points system. We must keep in mind that we were assessing the comprehensiveness of the model when making the prediction and not each variable separately (*Ramírez-Prado et al., 2015b*; *Palazón-Bru et al., 2016*; *Piqueras-Rodríguez et al., 2016*). On the other hand, variables that others have shown to be associated with poorer PPI adherence were not included (*Dal-Paz et al., 2012*; *Lanas et al., 2012*). However, without the use of these factors we obtained a very satisfactory discriminating capacity (AUC = 0.87). For example, if we had included the length of time of prescribed treatment it is likely that our AUC would have improved. Nonetheless, a value close to 90% is an indicator of high discrimination between the nonadherent and the adherent patient. Finally, as a methodological limitation, using a mobile application may have the disadvantage that some people may not be familiar with new technologies. However, these people could still apply the scoring system manually.

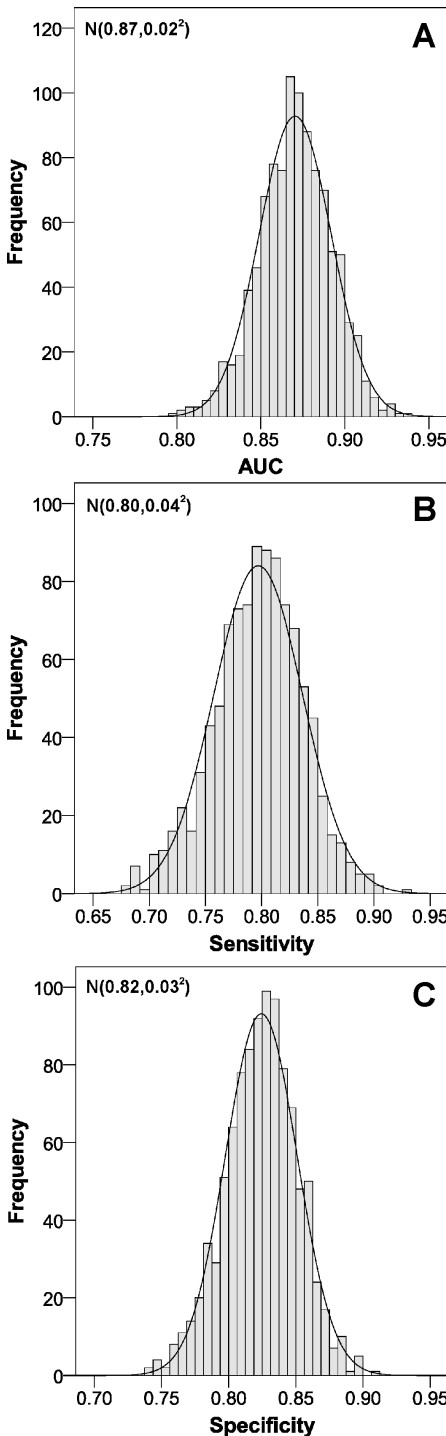

**Figure 4   Internal validation of the scoring system to predict nonadherence to proton pump inhibitors.** (A) area under the ROC curve; (B) sensitivity; (C) specificity. Abbreviations: AUC, area under the ROC curve.

## Comparison with the existing literature

If we compare our points system with the other methods for evaluating therapeutic nonadherence, we see that our system contains only objective variables, assessment is made in a matter of seconds (mobile application) and the estimation error is low. Pill count and days covered are complex to carry out owing to the need to verify whether the patient is taking the medication by checking the PPI container, which presents serious difficulties in routine clinical practice, since we cannot have the patient come in for an appointment only to see if he or she is taking the prescribed medication. On the other hand, the therapeutic nonadherence questionnaires require a good doctor-patient relationship and they are not as accurate as objective methods (*Perseguer-Torregrosa et al., 2014*). In other words, we have created an indirect method to assess nonadherence based on completely objective variables and that does not require a close doctor-patient relationship, because simply by looking at the prescription made by the doctor, we will be able to apply the prediction model.

Regarding the prevalence of nonadherence and its associated factors, our results were consistent with those found by others (*Dal-Paz et al., 2012*; *Lanas et al., 2012*; *Hedberg et al., 2013*; *Jonasson et al., 2013*; *Domingues & Moraes-Filho, 2014*; *Henriksson, From & Stratelis, 2014*), with the exception of antidepressant treatment, which was not assessed by others. We believe that this could be because a depressed person forgets to take medication. However, as other authors have suggested (*Martínez-St John et al., 2015*; *Rico-Ferreira et al., 2015*), this hypothesis should be analyzed through a qualitative study with this type of patient to determine the causes of PPI nonadherence.

## Implications to research and practice

External validation in other geographical areas in which the discriminatory capacity of the constructed test (AUC) is determined, as well as the sensitivity and specificity, is proposed. If the results obtained are similar to those of our study, we will have a screening test to detect which patient is not adhering to the PPI treatment and empathetically try to explain the consequences of nonadherence (*Dal-Paz et al., 2012*; *Jonasson et al., 2013*; *Domingues & Moraes-Filho, 2014*). Finally, it would be interesting to replicate this study with a longer follow-up time.

As the screening test contains objective parameters that can be obtained through the electronic prescription it is possible to determine at the pharmacy which patients are most at risk of PPI nonadherence and thus improve the relationship between the pharmacies and doctors, all to the benefit of the patient.

## CONCLUSIONS

A points system to predict nonadherence to PPIs has been constructed and internally validated. The system has been implemented in an application for Android. External validation of our prediction model in other geographical areas is planned. If similar results are obtained, we will have a screening tool to detect nonadherence and thus reduce possible gastrointestinal complications.

## ACKNOWLEDGEMENTS

The authors thank Maria Repice and Ian Johnstone for their help with the English language version of the text.

### Funding
The authors received no funding for this work.

### Competing Interests
Antonio Palazón-Bru is an Academic Editor for PeerJ.

### Author Contributions
- Emma Mares-García conceived and designed the experiments, performed the experiments, wrote the paper, reviewed drafts of the paper.
- Antonio Palazón-Bru conceived and designed the experiments, analyzed the data, wrote the paper, prepared figures and/or tables, reviewed drafts of the paper.
- David Manuel Folgado-de la Rosa conceived and designed the experiments, contributed reagents/materials/analysis tools, reviewed drafts of the paper, developed the mobile application.
- Avelino Pereira-Expósito, Álvaro Martínez-Martín, Ernesto Cortés-Castell and Vicente Francisco Gil-Guillén conceived and designed the experiments, contributed reagents/materials/analysis tools, reviewed drafts of the paper.

### Human Ethics

The following information was supplied relating to ethical approvals (i.e., approving body and any reference numbers):

The study was approved by the Ethics Committee of the Department of Health of Elda and both data collection and analysis were conducted anonymously and encrypted. All patients gave their consent to participate in the study orally.

### Data Availability

The raw data has been supplied as Data S1.

### Supplemental Information

Supplemental information for this article can be found online at http://dx.doi.org/10.7717/peerj.3455#supplemental-information.

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
