# Peer review of "Construction, internal validation and implementation in a mobile application of a scoring system to predict nonadherence to proton pump inhibitors"

_PeerJ, doi:10.7717/peerj.3455_

## Round 0.1 · original submission · Major Revisions

· Academic Editor

Major Revisions

We thank you for your submission of this manuscript at PeerJ. The three reviewers have highlighted a number of issues with the initial submission of this manuscript that require careful attention. I would therefore suggest you carefully read these comments and consider how you can improve the manuscript to the standard required by the three reviewers.

We suggest you have a fluent, preferably native, English-language speaker thoroughly copyedit your manuscript for language usage, spelling, and grammar. If you do not know anyone who can do this, you may wish to consider employing a professional scientific editing service for example American Journal Experts (AJE).

Best Regards

·

Basic reporting

Abstract, I suggest that the objective begin directly with the verb in infinitive, the other detail about the absence of study of this type and therefore the novelty of the research can be discussed later in the text.
Concise and sufficient introduction. Fairly clear. Language is correctly understood, with sufficient literature references.

Experimental design

The methodology followed is correct, the source of the sample is exhaustively described. Excessive text in the part of the methodology. I suggest in the section "Study desing and participants" summarizing the information in a figure where all phases of research are collected in this way the information will be more visual and easier to understand and to reproduce. In the section "Variables and meauserements" I also suggest summarizing or providing the information more clearly.
It would remove information from lines 175-183, only explain which tests have been performed for internal validation, remove the definitions provided in those lines.

Validity of the findings

It is a very interesting topic that research since the results obtained on this type of study can bring benefits to the community. The results of this research help increase the body of knowledge, in addition to being in relation to the current world with which we are developing, technology, with the creation of an application for the mobile.The conclusions give a response to the objective

Reviewer 2 ·

Basic reporting

The English language should be improve, You use term and paragraph unclear and with mistake in the gramatical form
The literature references are correct and current
The background is enough to present the idea of paper
In regarding to the tables and graph are correct

Experimental design

The design of the study correct although the explain of method is not clear. I don't understand all process and they do not have a high technical standard
You must show the number of Ethic Committee in the document
What software did you use to sample size? I can not see it

Validity of the findings

The analysis statistic is correct although I have not understood if the paper is to analysis of the predict nonadherence to proton pump inhibitors or the use of app to measure this

Additional comments

Line 128-140 This is discussion not method
The author use a lot of references in only one paragraphs , please remove references and use the most current
We do not have a paragraph of limitation of study in the discussion

·

Basic reporting

The title and subject of research are interesting as it is a novel idea . I have no problem with references and i think writting of the paper is adequate and clear in scientific terms. Reference and structure of the table and figures are also adequate

Experimental design

The objectives are adequately described in line 142 to 144.

As a methodological limitation, using a mobile application may have the disadvantage that some people are not related to new technologies.

On line 84 in relation to the study population and their follow-up refer to a prospective study, However I consider that 5 months of follow-up in total with 3 reviews may be a limitation.

Validity of the findings

In line 109 and up to 140 properly describe the procedures for selecting study variables that are relevant for the analysis of the data obtained in lines 144 to 146 perform an adequate calculation of the sample size using precise scientific methodology lines 142 a 147
Regarding statistics, the use of a logistic regression model is adequate for the events studied and the treatment of the study variables

The authors choose’s ROC curves and the establishment of cut-off points for classification is correct as a statistical method

The "Discussion" refers to the positive implications for practice with regard to the existing methods

Publishable work, but recommending extending the period of study. Interesting. It´s a new type of trend research using new technoligies whith clinical aplicatio

Additional comments

1. The title and subject of research are interesting as it is a novel idea due to the use of new technologies as a tool to know the adherence of treatment instead of traditional methods its new
2. The objectives are adequately described in line 142 to 144. The objective of our work was to determine if a points system correctly discriminated between patients who would present therapeutic nonadherence to PPIs and those who would not, we estimated the area under the ROC curve (AUC) of the points system.
4. Material and Methods. As a methodological limitation, using a mobile application may have the disadvantage that some people are not related to new technologies. (Line 67) Positively suggest other measurement sources that could skew measurement results such as measuring counting tablets .
5. On line 84 in relation to the study population and their follow-up refer to a prospective study, However I consider that 5 months of follow-up in total with 3 reviews may be a limitation.
6. In line 109 and up to 140 properly describe the procedures for selecting study variables that are relevant for the analysis of the data obtained in lines 144 to 146 perform an adequate calculation of the sample size using precise scientific methodology lines 142 a 147
7. Regarding statistics, the use of a logistic regression model is adequate for the events studied and the treatment of the study variables. (Lines 154-156) A binary logistic regression model was constructed in which the dependent variable was nonadherence to PPIs and The independent variables were our secondary variables (gender, age, antidepressants, omeprazole, NGRP of PPIs, and the total number of drugs)
8. The authors choose’s ROC curves and the establishment of cut-off points for classification is correct as a statistical method Lines 165-169 After calculating the score associated with each patient, the ROC curve was calculated and the cut-off point was established as The one that minimized the square root of (1-Sensitivity) 2+ (1-Specificity) 2. After obtaining this point, sensitivity, specificity, positive (PPV) and negative predictive values (NPV) and positive (PLR) and negative likelihood ratios (NLR) were calculated
9. Lines 177-183 Discrimination: a higher score can distinguish between the patient who does not adhere to treatment and the patient who does (assessed by calculating AUC); 2) Calibration: the probabilities of nonadherence to PPIs given by the model corresponding to those observed in reality (assessed by the sensitivity and the specificity). On the other hand, external validation consists of determining whether the predictive model is well calibrated and properly discriminated between adherent and nonadherent patients in a patient sample different to the one on which the model has been constructed
10. The writting of the results is adequate and clear in terms of the references it makes of sensitivity and specificity to classify study subjects as well as their representation in the article in Figures 1 and 2
11. The "Discussion" refers to the positive implications for practice with regard to the existing methods line 279 to 283 External validation in other geographical areas in which the discriminatory capacity of theconstructed test (AUC) is determined, as well The sensitivity and specificity, is proposed. If theresults obtained are similar to those of our study, we will have a screening test to detect which patient is not adhering to the PPI treatment and empathetically try to explain the consequences of nonadherence Being a positive aspect that has redudico bias and that has been properly Validated and could lead to other similar studies
12. Publishable work, but recommending extending the period of study. Interesting. It´s a new type of trend research using new technoligies whith clinical aplication

---

## Round 0.2 · accepted · Accept

· Academic Editor

Accept

We thank you for attending to the reviewers comments on your revised manuscript and am happy to let you know the paper has now been accepted for publication.

Reviewer 2 ·

Basic reporting

no comment

Experimental design

no comment

Validity of the findings

no comment

Additional comments

All questions and recommendations previously addressed have been corrected.

·

Basic reporting

The title and subject of research are interesting as it is a novel idea due to the use of new technologies as a tool to know the adherence of treatment instead of traditional methods its new

Experimental design

The objectives are adequately described

Material and Methods. As a methodological limitation, using a mobile application may have the disadvantage that some people are not related to new technologies.

Validity of the findings

Regarding statistics, the use of a logistic regression model is adequate for the events studied and the treatment of the study variables

The authors choose’s ROC curves and the establishment of cut-off points for classification is correct as a statistical method

The writting of the results is adequate and clear in terms of the references it makes of sensitivity and specificity to classify study subjects as well as their representation in the article in Figures 1 and 2

Additional comments

The changes introduced make the article publishable